# Non-Isothermal Crystallization Kinetics of Polyether-Ether-Ketone Nanocomposites and Analysis of the Mechanical and Electrical Conductivity Performance

**DOI:** 10.3390/polym14214623

**Published:** 2022-10-31

**Authors:** Xin Ye, Zhonglue Hu, Xiping Li, Sisi Wang, Jietai Ding, Mengjia Li, Yuan Zhao

**Affiliations:** Key Laboratory of Urban Rail Transit Intelligent Operation and Maintenance Technology & Equipment of Zhejiang Province, Zhejiang Normal University, Jinhua 321004, China

**Keywords:** PEEK/CNTs composites, crystallization behavior, electrical conductivity, electromagnetic interference shielding performance, annealing treatment

## Abstract

High-performance polyether-ether-ketone (PEEK) is highly desirable for a plethora of engineering applications. The incorporation of conductive carbon nanotubes (CNTs) into PEEK can impart electrical conductivity to the otherwise non-conductive matrix, which can further expand the application realm for PEEK composites. However, a number of physical properties, which are central to the functionalities of the composite, are affected by the complex interplay of the crystallinity and presence of the nanofillers, such as CNTs. It is therefore of paramount importance to conduct an in-depth investigation to identify the process that optimizes the mechanical and electrical performance. In this work, PEEK/CNTs composites with different carbon nanotubes (CNTs) content ranging from 0.5 to 10.0 wt% are prepared by a parallel twin-screw extruder. The effects of CNTs content and annealing treatment on the crystallization behavior, mechanical properties and electrical conductivity of the PEEK/CNTs composites are investigated in detail. A non-isothermal crystallization kinetics test reveals a substantial loss in the composites’ crystallinity with the increased CNTs content. On the other hand, mechanical tests show that with 5.0 wt% CNTs content, the tensile strength reaches a maximum at 118.2 MPa, which amounts to a rise of 30.3% compared with the neat PEEK sample after annealing treatment. However, additional annealing treatment decreases the electrical conductivity as well as EMI shielding performance. Such a decrease is mainly attributed to the relatively small crystal size of PEEK, which excludes the conductive fillers to the boundaries and disrupts the otherwise conductive networks.

## 1. Introduction

As a semi-crystalline, high-performance thermoplastic polymer, polyether-ether-ketone (PEEK) possesses a stimulating combination of properties, including high strength, light weight, superior chemical stability, high-temperature resistance, etc. These characteristics are highly desirable for a number of weight-sensitive, performance-demanding applications, such as aerospace, automobiles, constructions, etc., [1]. To fully leverage the superior properties of the PEEK matrix, the functionalities can be further tailored by incorporating with high-performance fibers. So far, a number of reinforcement fibers have been utilized to increase the intrinsic strength of the PEEK matrix, including carbon fiber (CF) [2,3], carbon black (CB) [4], boron nitride (BN) [5,6], carbon nanotubes (CNTs) [7,8], etc. Of all the fillers under consideration, CNTs have attracted significant attention due to their low density, high tensile strength and elastic modulus. For instance, Ata et al. [9] investigated the mechanical properties of CNT-reinforced PEEK. At 5.0 wt% CNT loading, the tensile strength of the composite increased to 112.0 MPa, a remarkable 24.0% increase from the neat PEEK (~90 MPa). In addition to its outstanding mechanical properties, CNT also exhibits exceptionally high electrical conductivity as well as aspect ratios, which are highly conducive for forming conductive networks within the insulating polymer matrix [10,11]. As a result, CNTs are among the most effective fillers for turning PEEK into highly functional conductive composites, endowing polymers with electrical conductivity at a rather low percolation threshold. Mohiuddin and Hoa [10] were among the earliest to evaluate the electrical conductivity of PEEK/CNT nanocomposites. With 10.0 wt% CNT, the electrical conductivity of the compression-molded composites reached 1.27 × 10^−5^ S/cm. A recent study by Li et al. [12] showed that, after incorporating 10.0 wt% CNTs in the PEEK matrix via melt mixing, the volume resistivity decreased considerably from 10^16^ Ω·cm to 11 Ω·cm.

Apart from tuning the mechanical strength and electrical conductivity, the presence of CNTs also plays a significant role in affecting the crystallization behavior of PEEK/CNT composites, which have attracted substantial interest from the research community. The CNTs typically act as nanoscale heterogeneous nucleators for the polymers and hence significantly enhance crystallization. However, studies on the effect of CNTs on the crystallization behaviors of PEEK have yielded inconsistent findings. Gohn et al. [13] discovered that the incorporation of CNT enhances nucleation at all cooling rates, resulting in an increase in the crystallinity with increased CNT content. The highest crystallinity reached 45.2% with 10.0 wt% CNTs when the composites were cooled at 5.0 °C/min to 400 °C, whereas the neat PEEK had a crystallinity of 37.5% under the same conditions. In contrast, Ata et al. [9] reported that the crystallinity only slightly decreased from 34.4 to 33.0% after incorporating 5.0 wt% CNTs within the PEEK matrix. Similar studies have been reported by Rinaldi et al. [14], who saw no significant difference in the degree of crystallinity between PEEK and PEEK/CNT composites.

On the other hand, the intrinsic crystallinity of PEEK is integral to the PEEK’s strength and electrical conductivity. When subjecting the semi-crystalline PEEK beyond its glass transition temperature, the polymer chains begin to rearrange and hitherto initiate the crystallization process, which has been found to be highly effective at enhancing strength and stiffness. For instance, Li et al. [15] revealed that, after keeping the neat PEEK film at 210 °C for a prolonged period of time, the overall crystallinity reached 29.7%, and the resultant strength reached 140.0 MPa, almost a 25.0% increase from the amorphous PEEK matrix. Crystallization also promotes electrical conductivity through expelling the conductive fillers to the boundaries of crystals, as dictated by the classic volume percolation theory [16]. Despite the vast amount of research performed on the interactions between the CNTs and PEEK, very few studies have recognized the synergic effects of polymer’s crystallization behavior and CNTs on the mechanical and electrical properties of PEEK. To bridge such a knowledge gap and to gain a better control on the physical properties of PEEK/CNTs composites, an in-depth study on the interactions between the heat treatment and CNTs is, therefore, highly desirable.

In this study, the effect of crystallization and CNTs on the mechanical and electrical properties of PEEK/CNTs composites are analyzed. The PEEK/CNTs composite was first prepared by melt-mixing and inject-molded into testing samples. Next, the effect of CNTs content on the rheology and crystallization behavior of the composites were characterized in detail.

## 2. Experimental

### 2.1. Materials

PEEK pellets (8200 G) with a density of 1.32 g/cm^3^ were acquired from Zhejiang PFLUON Chemical Co., Ltd. (Jinhua, China). The CNTs with an average diameter of 4.0~10.0 nm and length of 5.0~20.0 μm were also acquired by Zhejiang PFLUON Chemical Co., Ltd. (Jinhua, China).

### 2.2. Fabrication of the PEEK/CNTs Composites

The preparation process for the PEEK/CNTs composites is schematically shown in Figure 1a. Before melt-mixing, the raw PEEK pellets were dried at 120 °C for 4.0 h to completely remove the moisture. Then, the CNTs were mechanically mixed with PEEK in different amounts, varying from 0.5 to 10.0 wt%. Then, the mixed PEEK and CNTs were fed into a parallel twin-screw extruder (HAAKE PROCESS 16, Thermo Fisher, Waltham, MA, USA) and subsequently pelletized by a rotating pelletizer. After harvesting the PEEK/CNTs composite pellets, the composite tensile specimens were directly injection-molded by a micro-injection-molding machine (HAAKE MiniJet II, Thermo Fisher, Waltham, MA, USA); the injection-molding parameters are presented in Table 1. The geometry of the prepared tensile specimen followed the ASTM D638 standard (Type Ⅳ, as shown schematically in Figure 1b). To minimize the measurement error, at least five specimens were compression-molded for each composite. However, it is worth noting that, because of the extremely high viscosity with the high CNTs content, only the PEEK composites with below 5.0 wt% CNTs loading can be successfully injection-molded. To quantify the electrical conductivity and electromagnetic interference (EMI) shielding performance, the PEEK-CNTs composite was injection-molded into a rectangular specimen (length × width × thickness: 10 × 4 × 0.5, in mm) as shown schematically in Figure 1c.

### 2.3. Characterization

To reveal the dispersion of CNTs in the composites, the injection-molded specimens were cryo-fractured and subsequently characterized by a scanning electron microscope (EM-30PLUS, Coxem, Daejeon, Korea). The mechanical properties of PEEK/CNTs composites were tested by the universal material testing machine (UTM4204, Suns, Shenzhen, China) with a tensile speed of 20.0 mm/min.

To analyze the effects of CNTs content on the crystallization behavior of composites, the composite specimens were examined by differential scanning calorimetry (DSC 3500, Netzsch, Selb, Germany), where specimens were heated to 400 °C with a heating rate of 10 °C/min under a nitrogen atmosphere. After keeping the specimens at such a temperature for 5.0 min to eliminate thermal history, they were again cooled to room temperature at 30 °C/min. The crystallinity of composites could be expressed as
(1)$$X_{c}=\frac{\int_{T_{0}}^{T_{\infty }}\left(\frac{dH_{c}}{dT}\right)dT}{\Delta H_{c}\times \left(1-w_{CNT}\right)}\times 100\%$$
where *T*_0_ and *T_∞_*, are the onset and complete crystallization temperatures, *dH_c_* is the endothermic enthalpy of melting, Δ*H_c_* is the 100.0% crystallization enthalpy—for PEEK, it is 130.0 J/g —and $w_{CNT}$ is the weight fraction of CNTs.

Further, to investigate the effect of annealing treatment on the crystallinity and other functional properties (i.e., mechanical, electrical and electromagnetic) of PEEK and PEEK/CNTs composites, the neat PEEK and PEEK/CNTs composites were annealed at 280 °C for 4.0 h before being slowly cooled to room temperature.

The rheological behavior of PEEK/CNTs composites was characterized by the dynamic shear rheometer (HAAKE MARS 60, Thermo Fisher, Waltham, MA, USA). In the rheological test, the testing temperature was set to 400 °C, the strain was set to 1.0%, the scanning frequency range was 0.1–100.0 s^−^^1^ and the gap between the parallel plates was 1.0 mm.

The electrical conductivity of the composites was quantified by a digital multimeter (DMM 4040, Tektronix, Beaverton, OR, USA). The conductivity was calculated using Equations (2) and (3):(2)σ=1/ρ
(3)ρ=RS/L
where σ is the electrical conductivity of composites, σ is the resistivity of composites, *R* is the resistance of composites, *S* is the cross-sectional area of composites and *L* is the length of composites.

The electromagnetic shielding performance was evaluated by a vector network analyzer (N5234A, Agilent, Santa Clara, CA, USA) in the frequency range of 18.0−26.5 GHz (K-band) at room temperature. The physical parameters for evaluating the electromagnetic shielding performance can be obtained based on Equations (4)–(8), including SE_total_ (SE_T_), SE_reflection_ (SE_R_), and SE_absorption_ (SE_A_).
(4)$$R={\left|S_{11}\right|}^{2}={\left|S_{22}\right|}^{2}$$
(5)$$T={\left|S_{21}\right|}^{2}={\left|S_{12}\right|}^{2}$$
(6)$$SE_{R}=-10\log\left(1-R\right)$$
(7)$$SE_{A}=-10\log\left(\frac{T}{1-R}\right)$$
(8)$$SE_{T}\approx SE_{R}+SE_{A}$$

## 3. Results and Discussion

### 3.1. Influence of CNTs Content on Rheological Behavior

The viscoelasticity of the polymer melt, which plays a critical role in the molding process and part quality, is dependent on the temperature, shear rate and material composition [17]. Figure 2 shows the rheological properties of PEEK/CNTs composites with different CNTs content at 380 °C and the frequency is from 0.5 to 100.0 rad/s. As can be seen from Figure 2a,b, in every composite, both the storage modulus (G’) and loss modulus (G’’) increase as the frequency (ω) increases. Furthermore, the increases in G’ and G’’ are more significant with the increasing CNTs content at the same ω, and it is obvious at the higher ω. This phenomenon is caused by the characteristic relaxation time increasing with the CNTs content [18]. Figure 2c shows that the complex viscosity (η*) of PEEK/CNTs composites drops rapidly with the increasing ω, showing an apparent shear-thinning behavior. This is due to the fact that the CNTs and polymer chains preferentially align along the shear direction under higher strain rate, which leads to the drop in viscosity [19]. In addition to the apparent shear-thinning, the value of η* increases with the increasing CNTs content. Such an increase in the viscosity indicates that the CNTs have formed an effective network structure inside the composites, which increases the viscosity of composites [20]. Such a dramatic increase in the viscosity will result in higher shear stress during the melt-mixing, which is conducive to forming a uniform dispersion of the CNTs in the PEEK matrix. Figure 2d plots the loss factor (tanδ) of the samples versus frequency (ω) at 380 °C to characterize the elastic response of the composites. It can be seen that the composites have a viscoelastic transition within the tested frequency range. The viscous to elastic transition point, which corresponds to the peak in the loss factor, shifts to a high frequency as the CNTs content increases. In addition, because the increase in the CNTs content can enhance the stiffness of the composites, the tanδ gradually decreases. Moreover, at a higher CNTs content (over 1.0 wt%), the tanδ values become almost less than 1.0, which can be attributed to the formation of the network structure by the CNTs–CNTs interaction and CNTs–polymer chain interactions [20]. The described viscoelastic property of the composites also plays an integral role in the feasibility of the recently emerged 3D printing technology [21]. With proper tuning of these properties, the PEEK-CNTs composite may emerge as an appealing candidate for 3D printing [22].

### 3.2. Micro Morphologies of the PEEK/CNTs Composites

It is known that the performance of composites is closely related to the dispersion of nano fillers. According to the rheological behavior test (Section 3.1), the composites exhibit high viscosity. This result indicates that the composites will be subjected to higher shear stresses during the melt-mixing process, which is conducive to the uniform distribution of CNTs in the PEEK/CNTs composites [23]. Figure 3 shows the SEM images of cryo-fractured surfaces of the PEEK-CNTs composites with the varying CNT content. In the SEM images, the CNTs appear as a bright and elongated strand, as highlighted in the yellow circle. It is clear from the SEM images that, as the content of CNTs increase (Figure 3b–f), the captured CNTs become increasingly pervasive. Meanwhile, the distribution of those nanoscale CNTs fillers appear to remain homogeneous through all the compositions, and no apparent agglomeration is observed. Such well-dispersed CNTs within the PEEK matrix are conducive to forming a pervasive conductive network and reinforce the otherwise relatively weak matrix, which is highly desirable to achieve good mechanical properties and electrical conductivity. In addition to the distribution of CNTs, the fractured surfaces also appear to be largely ductile throughout all the compositions. This is another indication of the absence of severe CNT agglomeration, which would otherwise induce substantial stress concentration upon shearing and lead to brittle fracture.

### 3.3. Influence of CNTs Content on Non-Isothermal of PEEK/CNTs Composites

It is well known that promoting the nucleation of polymers and hindering the movement of the polymer chain are important factors that affect the crystallization process of polymers. Figure 4 shows the crystallization behavior of PEEK/CNTs composites at a constant cooling rate of 30 °C/min. The crystallinity of composites can be calculated using Equation (1), and the non-isothermal crystallization parameters are summarized in Table 2. When the CNTs content increases from 0.5 to 10.0%, the crystallization onset temperatures (T_s_) of PEEK/CNTs composites shifts to a higher temperature, increasing from 292 to 300 °C. However, the crystallinity with increasing CNTs content decreases significantly from 27.7 to 16.6%, proving that the addition of CNTs hinders the movement of molecules to decrease the crystallinity. The current study is consistent with the results of Li et al. [24]. Furthermore, it is observed that the crystallization peak temperature (*T_p_*) of the composites increases with the increasing CNTs content, indicating that CNTs can promote nucleation to increase crystallization rates. This phenomenon has also been observed by many researchers [25,26]. In summary, the change in crystallinity can be observed clearly, which indicates that hindering crystal growth is dominated during the crystallization process.

Jeziomy [27] extended the Avrami method, which was applicable to isothermal crystallization to the case of non-isothermal crystallization. In fact, the non-isothermal crystallization process can be approximately considered as an isothermal process by modifying the rate constant. The double-logarithmic form of the Avrami equation is shown in Equation (9). Compared with the isothermal crystallization process, the non-isothermal crystallization process is affected by the cooling rate. Therefore, the rate constant Z needs to be corrected by the cooling rates. The modified parameter for crystallization kinetics is given by Equation (10).
(9)$$\ln\left[-\ln\left(1-X_{t}\right)\right]=\ln Z+n\ln t$$
(10)$$\ln Z_{c}=\ln Z/\varphi$$

By using Equations (9) and (10), a plot of *ln(−ln(1−X_t_))* versus *lnt* gives the slope Avrami exponent *n* and the intercept lnZ, as shown in Figure 5. The linearity of the plots shows that the adaptation of the modified Avrami method for neat PEEK in the crystallization process is satisfactory. According to the kinetic parameters shown in Table 3, the Avrami exponent is not an integer. It can be observed that the *n* is stable around 3 with increasing the CNTs content, indicating that the addition of CNT does not change the growth pattern of crystals. When the value of *n* decreases, it proves that the crystal growth is limited during the crystallization process and results in the simpler geometry of spherulites. As for the effect of the CNTs content on $Z_{c}$, the higher CNTs content shows lower $Z_{c}$ values. It means that the crystal growth rates decrease at a higher CNTs content, which hinders the crystallization. *T_p_* increases with increasing filler, suggesting that CNTs can promote the formation of nucleation sites. As stated above, the addition of CNTs can promote nucleation while hindering the crystals’ growth. Meanwhile, the hindering effects become more dominant with the increasing CNTs loading.

### 3.4. Influence of Annealing Treatment on Crystallization of PEEK/CNTs Composites

Recently, many studies have found that the annealing treatment of polymers can improve the crystallization by the van der Waals force between the fillers. The crystallization plays an important role in the conductivity or mechanical properties [28,29]. Figure 6 illustrates the DSC thermograms of the samples before and after annealing treatment. From to Figure 6a, the thermograms of all the unannealed samples show a single melting endotherm. With the CNTs content increases, the area of melting endotherm decreases gradually, which means that the addition of CNTs reduces the crystallinity of the composites. It can be clearly seen from Table 4 that the neat PEEK melting temperature (T_m_) and PEEK/CNTs composites are similar, about 337.7 °C, indicating that the crystal structure of PEEK/CNTs composites does not change [16]. Nevertheless, when the CNTs content reaches 10%, the crystallinity of the composites reduces from 27.3 to 16.8%. This is consistent with the above analysis of non-isothermal crystallization kinetics.

As shown in Figure 6b, it can be found that the annealed samples display the secondary melting peak, and the position of the secondary melting peak is different with the CNTs content. Table 5 summarizes the peak temperatures (T_m1_, T_m2_) of both the endotherms during the polymer-melting process. Compared with the annealing treatment temperature, the second peak temperature is approximately 5–20 °C higher. This indicates that the small and imperfect crystals are present during annealing treatment. In the meantime, with the increase in CNTs content, the secondary peak shifts to the primary endotherm, which indicates that CNTs hinder the movement of the PEEK chain greatly. It is worth noting that the changes in crystallinity of annealed samples are the same as those of unannealed ones, which decrease with the increase in CNTs. Compared with the crystallinity of unannealed samples, the crystallinity of the annealed samples is higher than that of unannealed samples, with an increase of 18.3%, as the PEEK chain gains more energy during the annealing process [30]. 

### 3.5. Influence of Annealing Treatment on Mechanical Properties of PEEK/CNTs Composites

Figure 7 shows the tensile strength, elastic modulus, and maximum strain of the PEEK composites with different CNTs content (up to 5.0 wt%). It can be seen from Figure 7a that the tensile strength increases by as much as 8.7% with 5.0 wt% CNTs loading. It means that the incorporation of CNTs is conducive to improving the mechanical properties of composites. Figure 7b compares the effect of CNTs on the elastic modulus of the tested samples. When CNTs loading increases from 0.5 to 5.0 wt%, the elastic modulus increases from 22.5% and up to 34.6% compared to the neat PEEK. Such concurrent increases in the tensile strength and elastic modulus with the increasing CNTs loading implies a good interfacial interaction between the CNTs and PEEK matrix. This finding also indicates that there is no obvious agglomeration between CNTs, which is confirmed by SEM images. It is worth noticing that the enhancement in the elastic modulus appears less significant with higher CNTs. This is probably due to the decrease in the crystallinity in the composites with higher CNTs loading, as the crystallinity drops from 27.7% to 16.6%, as was shown in the earlier section. Such a finding is also consistent with other studies [14]. Taking the effect of crystallinity into account, the improvement of the elastic modulus can be attributed to the reinforcing effect of the filler itself. Figure 7c shows the maximum strain of composites with the increasing CNTs loading. It is obvious that the maximum strain decreases from 105.3 to 46.1% as the CNTs increase to 5.0 wt%.

To further unveil the effect of annealing on the mechanical properties of the PEEK/CNT composites, Figure 7 also compares the tensile strength, elastic modulus and elongation of non-annealed and annealed specimens. After annealing treatment, both the tensile strength and elastic modulus increase monotonically compared with the non-annealed sample. For instance, when the CNTs content is 5.0 wt%, the tensile strength and elastic modulus reach 118.2 MPa and 3070.0 MPa, respectively, which is 19.9% or 19.5% higher than the unannealed sample. In contrast, annealing results in a significant drop in plasticity, as shown in Figure 7c. After annealing, the maximum elongation drops from 104.0% to 64.0% for the neat PEEK. The drop in plasticity is also pervasive for the composite specimen. The decrease in the plasticity after annealing is much anticipated because of the increased crystallinity, which hinders the connection [31]. Beyond the mechanical properties, a recent study also indicated that the annealing process can tune the surface roughness of the specimen, which may merit investigation in the future studies.

### 3.6. Influence of Annealing Treatment on EMI Properties of PEEK/CNTs Composites

As a highly conductive nano filler, CNTs can effectively endow high electrical conductivity to the otherwise insulting polymeric matrix [32,33]. Figure 8a shows the electrical conductivity of the PEEK/CNTs composites with different CNTs content. It can be found that the electrical conductivity of composites increases with an increase in the CNTs content. When the CNTs content is 0.5 wt%, the conductivity of the composites increases about 9.0 orders of magnitude compared with the neat PEEK matrix, reaching 1.5 × 10^−3^ S/mm. Moreover, with the CNTs content being further increased, the electrical conductivity of the composites reaches a higher value. When the CNTs content is 10 wt%, the conductivity reaches 2.3 S/cm, almost 100 times higher than that of the 0.5 wt% CNT-loading specimen. This means that CNTs have formed a relatively complete conductive network in the PEEK matrix [34].

In order to further the influence of CNTs on the conductive network of composites, the classical percolation theory was used to analyze the conductive network in the composites with regard to the critical concentration of fillers, as shown in Equations (11) and (12):(11)$$V_{CNTs}=\frac{\omega \cdot \rho_{PEEK}}{\omega \cdot \rho_{PEEK}+\left(1-\omega \right)\cdot \rho_{CNTs}}$$
(12)$$\sigma =\sigma_{0}\cdot {\left(V-V_{c}\right)}^{t}$$
where *V_CNTs_* is defined as the CNTs volume fraction, *ω* denotes the CNTs weight fraction, *ρ_CNTs_* and *ρ_PEEK_* represents the density of the CNTs and PEEK, *σ* is the resistivity of the composite, *σ*_0_ denotes the resistivity of CNTs, *V* is defined as the filler volume, *V_c_* represents the percolation threshold, and *t* is the critical exponent. The composition and density of varied composite are present in Table 6. To facilitate the analysis of the critical index, Equation (12) is revised as follows:(13)$$log\left(\sigma \right)=log\left(\sigma_{0}\right)+t\cdot log\left(V-V_{c}\right)$$
where the critical exponent *t* is transformed into the slope of the function, as shown in Figure 8.

Previous studies have shown that the percolation threshold of thermoplastics polymer is 1.0~5.0 wt%, and the percolation threshold of PEEK/CNTs composite system is usually 3.5 wt% [35]. From Figure 8, the critical exponent of PEEK/CNTs composites was determined to be 1.31 by ordinary least squares fitting. Due to the small data sample, the critical exponent may have a large error. Nonetheless, Weber and Kamal [36] reported the critical exponent as being between 1.3 and 3.1 for different polymer and reinforcement systems. It can be seen that the critical exponent of the PEEK/CNTs system is in reasonable agreement with the experimental and theoretical predictions.

To further reveal the effect of annealing on the electrical conductivity of PEEK/CNTs composites, the samples with different CNTs content were annealed at 280 °C for 4.0 h. Similarly, the electrical conductivity of thermal annealed PEEK/CNTs composites was measured, and the test results are shown in Figure 9a. It is worth noticing that, for all the composite specimens, the electrical conductivity decreased after annealing. For instance, after annealing, the electrical conductivity of PEEK/CNTs composites with 10.0 wt% CNTs reduced from 2.3 to 2.1 S/cm. This result, however, is contradictory to a few previous studies where annealing appears to boost the electrical conductivity [37]. One probable explanation for such a discrepancy is the extremely small-sized crystallites in PEEK. As shown in Figure 9b, the polarizing microscope of neat PEEK shows very small-sized crystallites. The incorporation of nanoscale CNTs enables the formation of a highly conductive network. After annealing treatment, however, the crystal size grew and the crystallinity of the composites increased, leading to the exclusion of CNTs to the boundaries of the crystals, thus disrupting the otherwise conductive network. Nevertheless, the conductivity drop is not significant in all the composites demonstrated in this study, as the volume exclusion effect can possibly bring new pathways for electric conductivity [16].

With such superior electrical conductivity, the as-prepared PEEK/CNTs composites possess great potential for a variety of functional applications, such as electromagnetic shielding performance. The ability of composites to resist the electromagnetic waves can be measured by electromagnetic shielding performance (SE). Usually, when the SE reaches 30.0 dB, it means that the composites can attenuate 99.9% of electromagnetic waves. Figure 10a shows the electromagnetic shielding properties of PEEK/CNTs composites with different CNTs content. As the CNTs content increased, the SE increased dramatically. When the CNTs loading was above 7.0 wt%, the composites exhibited outstanding shielding performance, with SE exceeding 30.0 dB. To further identify the components responsible for the shielding effect, Figure 10b shows the quantitative reflection loss (SE_R_) and absorption loss (SE_A_) for the PEEK/CNTs composites with varying CNTs. While both SE_A_ and SE_R_ increased monotonically with the increasing CNTs content, it is apparent that SE_A_ is the major component for the electromagnetic shielding. For instance, when the CNTs content reached 10.0 wt%, the SE_A_ reached 27.8 dB, which accounts for more than 80.0% of the SE_T_. This meant that when electromagnetic waves propagate through composites, higher conductivity causes more current to convert electromagnetic energy into heat, increasing the absorption loss [38]. In other words, the improvement of electromagnetic shielding properties by CNTs is mainly due to the increase in the absorption of incident electromagnetic waves, rather than increasing the surface reflection loss.

It is also interesting to investigate the effect of annealing on the electromagnetic shielding performance. After annealing all samples at 280 °C for 4.0 h, the EMI shielding performance is shown in Figure 10c. Compared with Figure 10a,c, the SE dropped significantly after annealing treatment. Such a drop is consistent with the change in the electrical conductivity. For example, the SE_T_ of the PEEK/CNTs composites with 10.0 wt% CNTs was 34.7 dB, but it dropped to 29.6 dB with a decrease of 14.7% after annealing treatment. In order to clearly investigate the changes in SE_A_, SE_R_ and SE_T_ after annealing, Figure 10d shows the variation of SE_A_, SE_R_ and SE_T_ at 22 GHz. Compared with the values of SE_R_, the values of SE_A_ dropped more obviously, decreasing about 15.6%. However, SE_A_ still accounted for 79.2% of SE_T_, indicating that the electromagnetic shielding properties of annealed PEEK/CNTs composites are still dominated by absorption loss. Earlier studies have indicated the EMI absorption loss is strongly related to the electrical conductivity of the tested specimen [39,40], which explains the drop in the EMI shielding performance after annealing.

## 4. Conclusions

In this paper, a series of PEEK/CNTs composites were prepared by melt-extrusion and subsequent injection-molding. The effects of CNTs content, annealing treatment on the crystallization, mechanical properties, electrical conductivity, and electromagnetic shielding properties were investigated in detail. The incorporation of CNTs reduced the overall crystallinity from 32.3 to 17.7%. On the other hand, the widespread and pervasive CNTs significantly enhanced the strength and the electrical conductivity of the PEEK matrix. Both the tensile strength and electrical conductivity of PEEK-CNTs composite increased significantly compared with the unfilled counterpart. The highly conductive PEEK-CNTs composites also exhibited outstanding EMI shielding performance, reaching 34.7 dB (with 10 wt% CNTs loading). Moreover, this study also reveals that the post-processing annealing treatment is highly effective in increasing the overall strength of the composites but is detrimental for the electrical conductivity and hence the EMI shielding performance, as the grown crystals disrupt the otherwise connected conductive networks. This study can hopefully spur new ideas on the composite design of PEEK-based composites and incentivize new strategies for performance optimization via heat treatment.

## Figures and Tables

**Figure 1 polymers-14-04623-f001:**
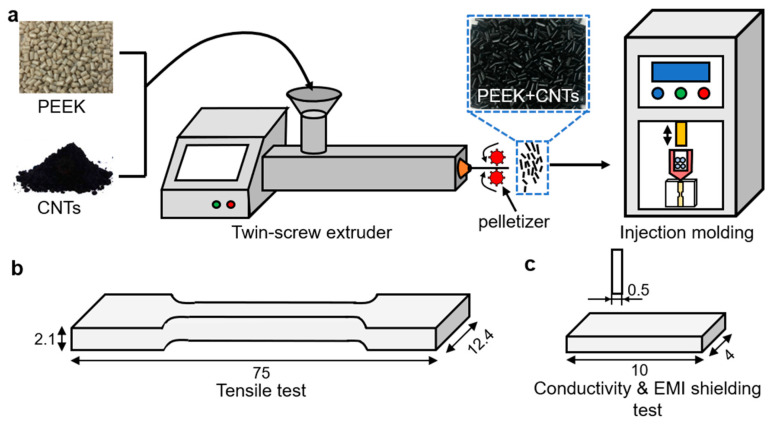
(**a**) Fabrication process for the PEEK/CNTs composites with a twin-screw extruder; (**b**) tensile specimen schematic; (**c**) conductive specimen schematic.

**Figure 2 polymers-14-04623-f002:**
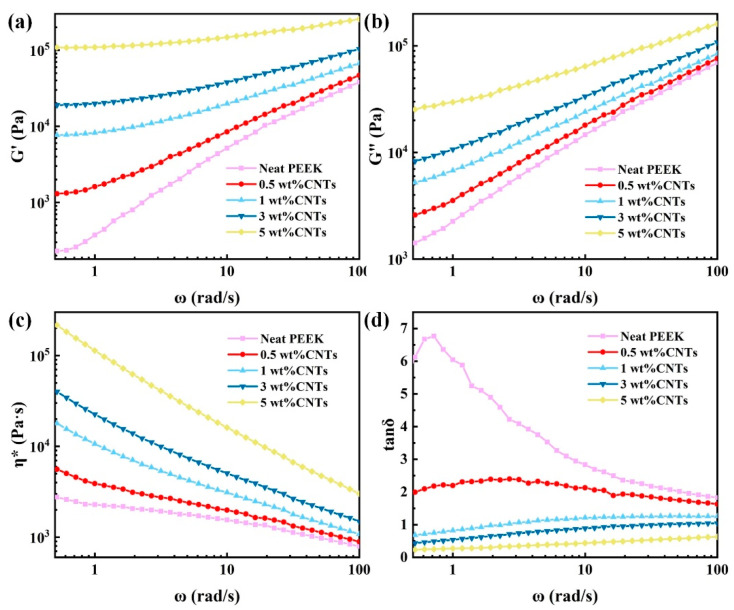
The viscoelastic behavior of PEEK/CNTs composites with different CNTs content: (**a**) the storage modulus (G’); (**b**) loss modulus (G’’); (**c**) complex viscosity (η*); and (**d**) loss factor (tanδ).

**Figure 3 polymers-14-04623-f003:**
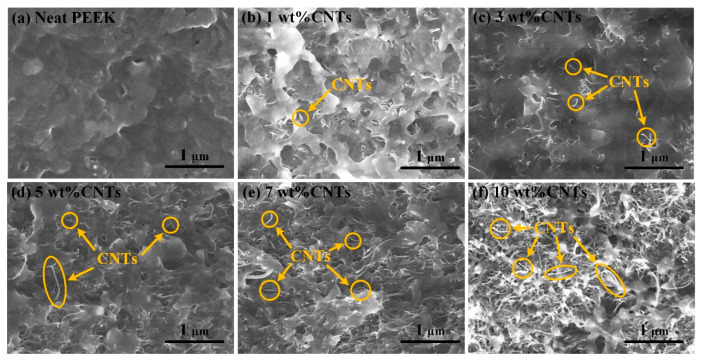
SEM images of PEEK/CNTs composite with different CNTs content. (**a**) Neat PEEK; (**b**) 1 wt% CNTs; (**c**) 3 wt% CNTs; (**d**) 5 wt% CNTs; (**e**) 7 wt% CNTs; (**f**) 10 wt% CNTs.

**Figure 4 polymers-14-04623-f004:**
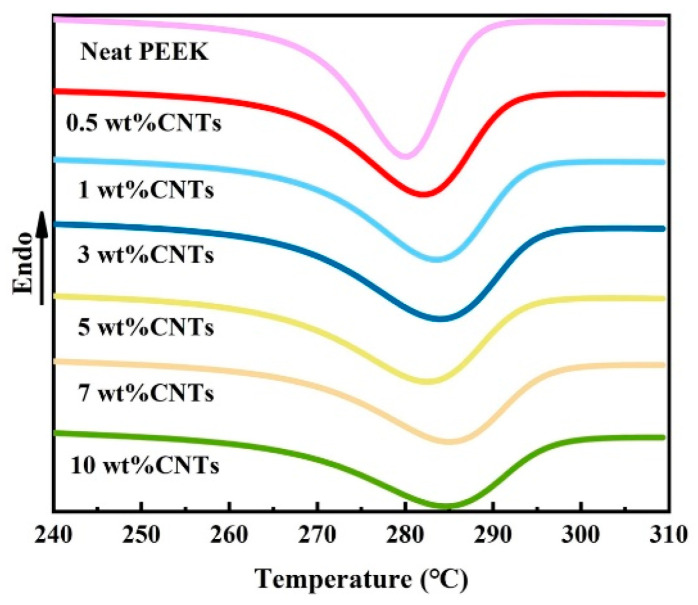
DSC cooling curves at various CNTs content for PEEK/CNTs composites.

**Figure 5 polymers-14-04623-f005:**
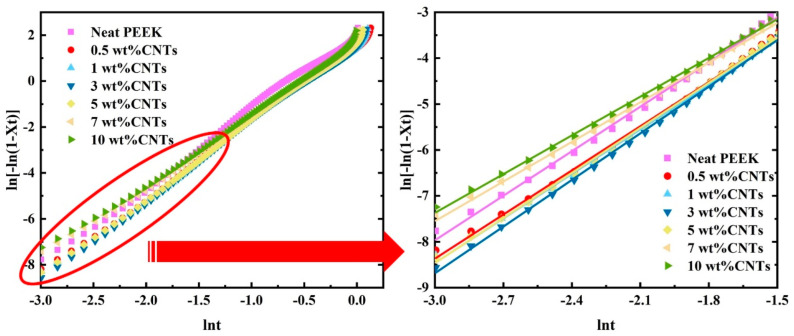
Avrami plots of ln(−ln(1−X_t_)) versus lnt at various cooling rates under non-isothermal crystallization.

**Figure 6 polymers-14-04623-f006:**
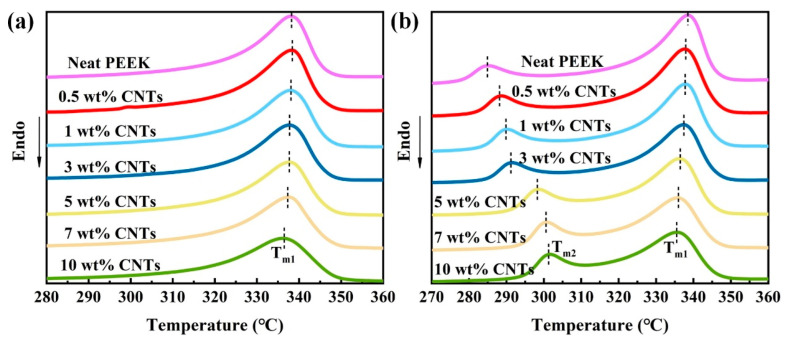
DSC heating curve at various CNTs content for PEEK/CNTs composites: (**a**) before annealing treatment, (**b**) after annealing treatment.

**Figure 7 polymers-14-04623-f007:**
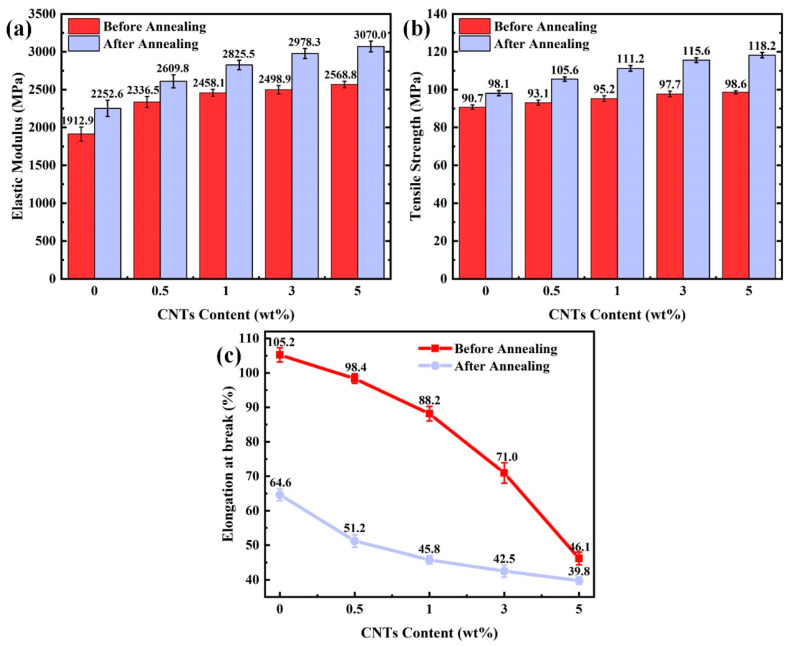
Mechanical properties of PEEK/CNT composites with different CNT content: (**a**) tensile strength; (**b**) elastic modulus; (**c**) elongation at break.

**Figure 8 polymers-14-04623-f008:**
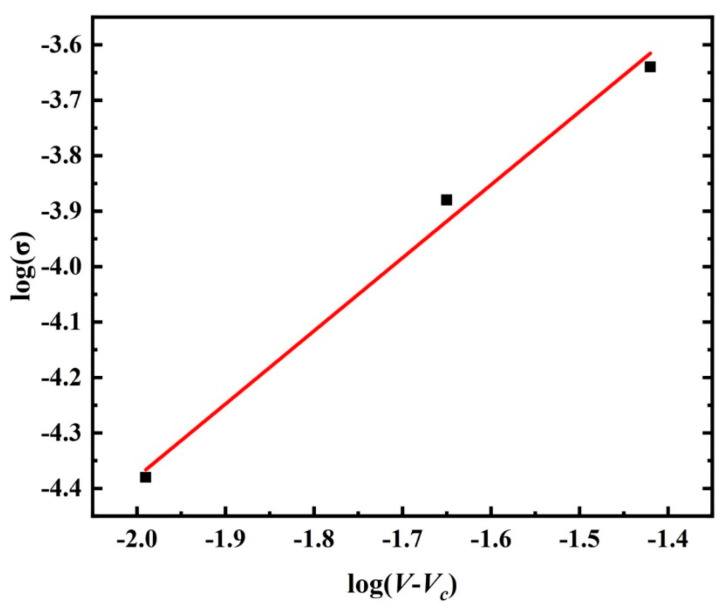
Classical percolation theory.

**Figure 9 polymers-14-04623-f009:**
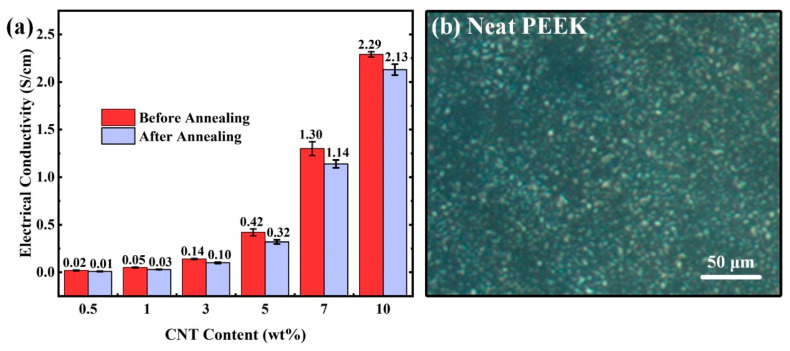
(**a**) Electrical conductivity of PEEK/CNTs composites with different CNTs content; (**b**) Polarizing optical microscope image for neat PEEK.

**Figure 10 polymers-14-04623-f010:**
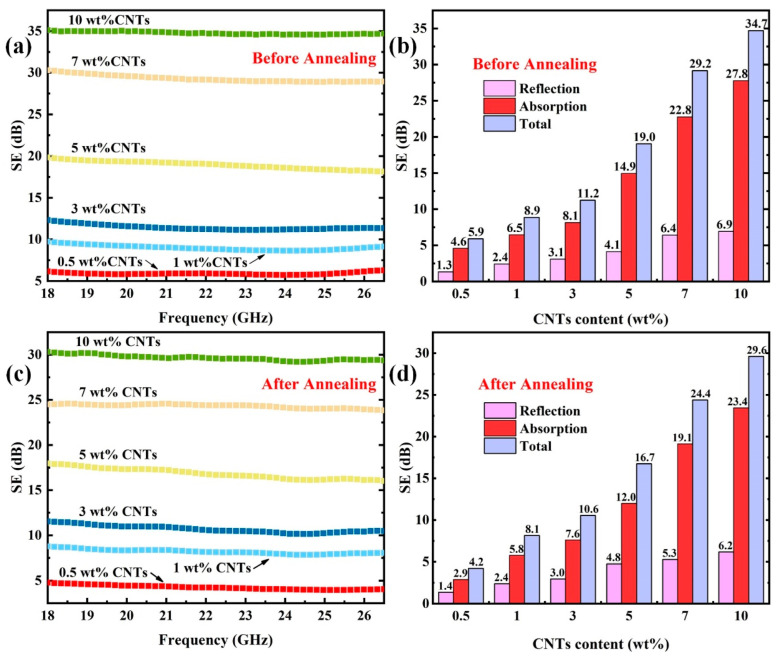
EMI shielding efficiency of PEEK/CNTs composites with different CNT contents in the range from 18 GHz to 26.5 GHz: (**a**) before annealing; (**c**) after annealing; the SE (SE_T_, SE_A_, SE_R_) for the PEEK/CNTs composites at the frequency of 22 GHz: (**b**) before annealing; (**d**) after annealing.

**Table 1 polymers-14-04623-t001:** Parameters for injection-molding.

Parameters	Value
Melt temperature (°C)	400
Mold temperature (°C)	200
Injection pressure (MPa)	80
Injection time (s)	5
Packing time (s)	10

**Table 2 polymers-14-04623-t002:** Non-isothermal crystallization parameter of the PEEK/CNTs composites.

CNTs Content (wt%)	T_s_ (°C)	*T_p_* (°C)	Crystallinity (%)
Neat PEEK	292.0	280.0	27.7
0.5	296.5	282.1	25.0
1.0	298.5	283.6	24.1
3.0	299.5	284.0	23.4
5.0	298.0	282.6	20.5
7.0	300.0	285.0	18.5
10.0	300.0	284.6	16.6

**Table 3 polymers-14-04623-t003:** Kinetics parameters of PEEK for various filler content.

CNTs Content (wt%)	*n*	Z	Z_c_
Neat PEEK	3.46	10.04	1.40
0.5	3.21	3.49	1.12
1.0	3.23	3.45	1.12
3.0	3.39	4.39	1.16
5.0	3.27	3.89	1.14
7.0	2.89	2.96	1.104
10.0	2.81	2.92	1.102

**Table 4 polymers-14-04623-t004:** Crystallization parameter of the PEEK/CNTs composites before annealing.

CNTs Content (wt%)	T_m_ (°C)	Crystallinity (%)
Neat PEEK	338.1	27.3
0.5	338.3	26.6
1.0	338.0	25.4
3.0	337.7	23.6
5.0	337.7	21.5
7.0	337.6	19.2
10.0	336.2	16.8

**Table 5 polymers-14-04623-t005:** Crystallization parameter of the PEEK/CNTs composites after annealing.

CNTs Content (wt%)	T_m1_ (°C)	T_m2_ (°C)	Crystallinity (%)
Neat PEEK	338.5	285.2	32.3
0.5	337.7	288.6	30.7
1.0	337.9	290.2	28.7
3.0	337.4	291.5	26.1
5.0	336.3	298.6	23.4
7.0	335.8	300.8	20.9
10.0	335.6	301.6	17.7

**Table 6 polymers-14-04623-t006:** CNTs volume fraction of varied composites.

CNTs Content (wt%)	CNTs Volume Fraction (%)
0.5	0.31
1.0	0.62
3.0	1.86
5.0	3.10
7.0	4.34
10.0	6.20

## Data Availability

The datasets generated and analyzed during the current study are available from the corresponding author upon reasonable request.

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
