# Peer review of "Non-Isothermal Crystallization Kinetics of Polyether-Ether-Ketone Nanocomposites and Analysis of the Mechanical and Electrical Conductivity Performance"

_polymers, 2022, doi:10.3390/polym14214623_

Round 1
Reviewer 1 Report
The work «Non-isothermal crystallization kinetics of polyether-ether-ketone nanocomposites and analysis of the mechanical and electrical conductivity performance» is devoted very actual topic, because the study of functional properties improvement by composite producing is wide-spreading, efficient and necessary for the correct selection of composite components. The technique for obtaining composites is well illustrated. Numerous parameters are investigated, various dependences are obtained and discussed. Interesting results presented in the form of graphs, photomicrographs and summary tables are shown. Article and conclusions are well structured. But I have some comments:
1. In section "2.3. Characterization" it is necessary to add information about the shape of samples for the study of electrical conductivity and electromagnetic shielding performance.
2. The electrical conductivity in PEEK/CNT samples is described by the percolation law. It is necessary to add an analysis of the percolation dependence of the conductivity on the CNT concentration.
3. In lines 390-391 "However, additional annealing treatment decreases the electrical conductivity as well as EMI shielding performance", this conclusion should be discussed in more detail in section "3.6. Influence of annealing treatment on EMI properties of PEEK/CNTs composites". Also in section 3.6 there is no comparison with known similar cnt peek composites.
Author Response
Response to Reviewer 1 Comments
We gratefully thank the editor and all reviewers for their constructive remarks and useful suggestions, which have significantly raised the quality of the manuscript and also allowed us to further improve the manuscript. We have made point-to-point responses as listed below. We hope our revisions can meet the satisfactory standard of the reviewers. The text changes we have made in the revised manuscript are highlighted in red color.
Reviewer 1: The work «Non-isothermal crystallization kinetics of polyether-ether-ketone nanocomposites and analysis of the mechanical and electrical conductivity performance» is devoted very actual topic, because the study of functional properties improvement by composite producing is wide-spreading, efficient and necessary for the correct selection of composite components. The technique for obtaining composites is well illustrated. Numerous parameters are investigated, various dependences are obtained and discussed. Interesting results presented in the form of graphs, photomicrographs and summary tables are shown. Article and conclusions are well structured. But I have some comments:
Our response: We gratefully appreciate for the reviewer’s overall positive comments on this study. To address the concerns raised by the reviewer, we have listed the point-to-point responses in the following.
Point 1: In section "2.3. Characterization" it is necessary to add information about the shape of samples for the study of electrical conductivity and electromagnetic shielding performance.
Response 1: We thank reviewer for raising this question. For the both electrical conductivity and electromagnetic shielding test, the test samples were injection-molded into a rectangular shape (length × width × thickness: 10×4×0.5 mm), as shown schematically in Figure 1c. To clarify, the text in the manuscript has been revised as:
“To quantify the electrical conductivity and electromagnetic shielding performance, the PEEK-CNTs composite was injection-molded into a rectangular specimen (length × width × thickness: 10×4×0.5, in mm), as shown schematically in Figure 1(c).”
The notation in Figure 1(c) has also been updated accordingly.
Point 2: The electrical conductivity in PEEK/CNT samples is described by the percolation law. It is necessary to add an analysis of the percolation dependence of the conductivity on the CNT concentration.
Response 2: We thank the reviewer for pointing it out. We agree with the reviewer that the electrical conductivity in PEEK/CNT samples can be described by the percolation law, where the effectives electrical conductivity is described by a power-law function, as following:
In order to better further the influence of CNTs on the conductive network of composites, the classical percolation theory was used to analyze the conductive network in the composites with regard to the critical concentration of fillers, as shown in Eq. (11) and Eq. (12):
|
|
(11) |
|
|
|
(12) |
Where VCNTs is defined as CNTs volume fraction, ω denotes the CNTs weight fraction, ρCNTs and ρPEEK represents the density of CNTs and PEEK, σ is the resistivity of composite, σ0 denotes resistivity of CNTs, V is defined as filler volume, Vc represents the percolation threshold, t is the critical exponent. The composition and density of varied composite are present in Table 6. To facilitate the analysis of the critical index, Eq. (12) is revised as follows:
|
|
(13) |
Where the critical exponent t will be transformed into the slope of the function, as shown in Figure 8.
Figure 8. Classical percolation theory
Previous studies have shown that the percolation threshold of thermoplastics polymer is between 1.0 ~ 5.0 wt%, and the percolation threshold of PEEK/CNTs composite system is usually 3.5 wt%[28]. From Figure 8, the critical exponent of PEEK/CNTs composites was determined to be 1.31 by ordinary least squares fitting. Due to the small data sample, the critical exponent may have a large error. Nonetheless, Weber and Kamal[29] reported the critical exponent between 1.3 and 3.1 for different polymer and reinforcement systems. It can be seen that the critical exponent of the PEEK/CNTs system is in reasonable agreement with the experimental and theoretical prediction.
Table 6. CNTs volume fraction of varied composites
|
CNTs content (wt%) |
CNTs volume fraction (%) |
|
0.5 |
0.31 |
|
1.0 |
0.62 |
|
3.0 |
1.86 |
|
5.0 |
3.10 |
|
7.0 |
4.34 |
|
10.0 |
6.20 |
Point 3: In lines 390-391 "However, additional annealing treatment decreases the electrical conductivity as well as EMI shielding performance", this conclusion should be discussed in more detail in section "3.6. Influence of annealing treatment on EMI properties of PEEK/CNTs composites". Also in section 3.6 there is no comparison with known similar cnt peek composites.
Response 3: We thank the reviewer for raising this concern. The adverse effect of annealing treatment on the electrical conductivity and electromagnetic shielding has been discussed in the section 3.6 (line 318-322). Moreover, relevant references are added to prove our point and the references have also been updated accordingly.
Reviewer 2 Report
Dear Authors,
The article is very interesting, I recommend few changes:
1. In the text you mentioned the importance of viscoelastic properties. These materials can be also used in 3D printing, so in my opinion, it would be good to relate to the topic e.g. in publication :
Viscoelastic Properties of Cell Structures Manufactured Using a Photo-Curable Additive Technology-PJM,
DOI b10.3390/polym13111895
2. I think that point 3.2. Micro morphologies of the PEEK/CNTs composites
should be better in depth describe in the text in point 3.2, especially figure 3 and its analysis.
3. How do you explain the deviation in the initial stage shown in figure 5 in the red line?
4. In point 3.4 you relate to thermal treatment for plastic, I think that interesting results which can improve the introduction and relate also to figure 7 in this area are presented in the publication:
Quality of the Surface Texture and Mechanical Properties of FDM Printed Samples after Thermal and Chemical Treatment
DOI
10.5545/sv-jme.2019.6322
5. I think that data can be removed from the conclusion to the discussion and described more results, especially about the thermal treatment results in conclusion which has very practical application
Regards,
Reviewer
Author Response
Response to Reviewer 2 comments
We gratefully thank the editor and all reviewers for their constructive remarks and useful suggestions, which have significantly raised the quality of the manuscript and also allowed us to further improve the manuscript. We have made point-to-point responses as listed below. We hope our revisions can meet the satisfactory standard of the reviewers. The text changes we have made in the revised manuscript are highlighted in red color.
Reviewer 2: The article is very interesting; I recommend few changes:
Our response: We gratefully appreciate for the reviewer’s positive comments on this study. The point-to-point responses have been listed in the following.
Point 1: In the text you mentioned the importance of viscoelastic properties. These materials can be also used in 3D printing, so in my opinion, it would be good to relate to the topic e.g. in publication: Viscoelastic Properties of Cell Structures Manufactured Using a Photo-Curable Additive Technology-PJM.
Response 1: We appreciate the reviewer for raising this point. Indeed, with appropriate viscoelastic properties, the presented PEEK-CNT composites are well-suited for the 3D printing, as demonstrated in one of our recent publications (X. Ye, et al, Addit. Manuf., 2022, 59, 103188).
Here, the manuscript has been revised as following:
…CNTs-polymer chain interactions. The described viscoelastic property of the composites also plays an integral role on the feasibility of the recently emerged 3D printing technology. With proper tuning of these properties, the PEEK-CNTs composite may emerge as an appealing candidate for 3D printing.
The references have also been revised accordingly.
Point 2: I think that point 3.2. Micro morphologies of the PEEK/CNTs composites should be better in depth describe in the text in point 3.2, especially figure 3 and its analysis.
Response 2: We appreciate the reviewer for pointing this out. We have reconstructed the major portion of the Section 3.2, and the manuscript has been revised as following:
… the PEEK/CNTs composite. Figure 3 shows the SEM images of cryo-fractured surfaces of the PEEK-CNTs composites with varying CNT content. In the SEM images, the CNTs appear as bright and elongated strand, as highlighted in the yellow circle. It is clear from the SEM images that, as the content of CNTs increase (Figure 3(b)-(f)), the captured CNTs become increasingly pervasive. Meanwhile, the distribution of those nanoscale CNTs fillers appear to remain homogeneous through all the compositions, and no apparent agglomeration was observed. Such well-dispersed CNTs within the PEEK matrix are conducive to form pervasive conductive network and reinforce the otherwise relatively-weak matrix, which are highly desirable to achieve good mechanical properties and electrical conductivities. In additional to the distribution of CNTs, the fractured surfaces also appear to be largely ductile throughout all the compositions. This is another indication on the absence of severe CNT agglomeration, which would otherwise induce substantial stress concentration upon shear and lead to brittle fracture.
Point 3: How do you explain the deviation in the initial stage shown in figure 5 in the red line?
Response 3: Due to the addition of CNTs, the high aspect ratio will hinder the growth of polymer crystals, destroy the integrity of the crystals, and lead to the gradual reduction of n. With the addition of CNTs, heterogeneous nucleation of composites was promoted and crystal growth was hindered. Among them, the hindrance plays a dominant role, leading to the Zc decreases. With the gradual increase of CNTs content, heterogeneous nucleation was gradually enhanced, resulting in a gradual increase in Zc. However, when the content of CNTs is further increased (> 5.0 wt%), the steric hindrance effect of CNTs on polymer molecular chains becomes more obvious, which reduces the mobility of molecular chain segments and the crystallization rate, leading to the decrease of Zc. Similar results have been demonstrated in many studies of crystallization kinetics, such as:
- Fu, et al, Non-isothermal crystallization kinetics of graphene/PA10T composites, 2022, ttps://doi.org/10.1016/j.heliyon.2022.e10206
- Yu, et al, Non-isothermal crystallization kinetics of poly(ether sulfone) functionalized graphene reinforced poly(ether ether ketone) composites, 2021, https://doi.org/10.1016/j.polymertesting.2021.107150
- Regis, et al, Opposite role of different carbon fiber reinforcements on the non-isothermal crystallization behavior of poly(etheretherketone), 2016, https://doi.org/10.1016/j.matchemphys.2016.05.034
- Kuo, et al, On the crystallization behavior of the nano-silica filled PEEK composites, 2010, https://doi.org/10.1016/j.matchemphys.2010.04.043
Point 4: In point 3.4 you relate to thermal treatment for plastic, I think that interesting results which can improve the introduction and relate also to figure 7 in this area are presented in the publication: Quality of the Surface Texture and Mechanical Properties of FDM Printed Samples after Thermal and Chemical Treatment. DOI: 10.5545/sv-jme.2019.6322
Response 4: We thank the reviewer for pointing this out. As suggested by the reviewer, the heat treatment not only tunes the properties of the composite (i.e. mechanical and electrical properties), but also have implications for composites processed with other techniques, such as 3D printing. Here, the manuscript has been revised as following (in Section 3.5):
… increased crystallinity, which hinders the connection. Beyond the mechanical properties, a recent study has also indicated the annealing process can tune the surface roughness of the specimen, which may merit further investigation in the future studies.
The references have also been updated accordingly.
Point 5: I think that data can be removed from the conclusion to the discussion and described more results, especially about the thermal treatment results in conclusion which has very practical application
Response 5: We thank the reviewer for pointing it out. We have revised the conclusion section, to make it more concise and also highlighted the effect of thermal treatment on the electrical and electromagnetic properties of the specimens. The new conclusion section now reads as following:
In this paper, a series of PEEK/CNTs composites were prepared by melt-extrusion and subsequent injection molding. The effects of CNTs content, annealing treatment on the crystallization, mechanical properties, electrical conductivity, and electromagnetic shielding properties were investigated in detail. The incorporation of CNTs reduces the overall crystallinity from 32.3% to 17.7%. On the other hand, the widespread and pervasive CNTs significantly enhances the strength and the electrical conductivity of the PEEK matrix. Both the tensile strength and electrical conductivity of PEEK-CNTs composite increases significantly compared with the unfilled counterpart. The highly conductive PEEK-CNTs composites also exhibits outstanding EMI shielding performance, reaching 34.7 dB (with 10wt% CNTs loading). Moreover, this study also reveals that, the post-processing annealing treatment is highly effective in increasing the overall strength of the composites, but is detrimental for the electrical conductivity and hence the EMI shielding performance as the grown crystals disrupts the otherwise connected conductive networks. This study can hopefully spur new ideas on composite design of PEEK-based composites and incentive new strategies for performance optimization via heat treatment.
Reviewer 3 Report
The manuscript titled: "Non-isothermal crystallization kinetics of polyether-ether-ketone nanocomposites and analysis of the mechanical and electrical conductivity performance", is a very interesting paper.
In this manuscript is presented a series of PEEK/CNTs composites that were obtained by melt mixing, with investigated effects of CNTs content, annealing treatment on the crystallization, mechanical properties, electrical conductivity and electromagnetic shielding properties. Although the authors presented a number of methods of characterization of the obtained PEEK/CNTs composites, the chemical composition was not presented. More precisely, not a single method was applied in order to prove the chemical composition of the samples. It is necessary for the authors to add the CNOHS analysis of the CNTs sample, the analysis of the composition of polyether-ether-ketone, the analysis of the composition of all obtained PEEK/CNTs composites, as well as the XRD analysis of PEEK/CNTs composites in order to prove the degree of crystallinity that they constantly mention throughout the manuscript and thus connect all the characterized properties with crystallinity.
In addition, despite all mentioned good properties of the PEEK/CNTs composites, it is necessary to specify the concrete application of the obtained composites with emphasis on the scientific contribution. This type of examination, without clearly definition of the scientific contribution and mechanisms of influence of reactants PEEK/CNTs composites, represents the technical report. I kindly ask authors to add a scientific contribution.
I recommend the accept after minor revision (corrections to minor methodological errors and text editing)
Author Response
Response to Reviewer 3 Comments
We gratefully thank the editor and all reviewers for their constructive remarks and useful suggestions, which have significantly raised the quality of the manuscript and also allowed us to further improve the manuscript. We have made point-to-point responses as listed below. We hope our revisions can meet the satisfactory standard of the reviewers. The text changes we have made in the revised manuscript are highlighted in red color.
Reviewer 3: The manuscript titled: "Non-isothermal crystallization kinetics of polyether-ether-ketone nanocomposites and analysis of the mechanical and electrical conductivity performance", is a very interesting paper.
Our response: We gratefully appreciate for the reviewer’s positive comments on this study.
In this manuscript is presented a series of PEEK/CNTs composites that were obtained by melt mixing, with investigated effects of CNTs content, annealing treatment on the crystallization, mechanical properties, electrical conductivity and electromagnetic shielding properties. Although the authors presented a number of methods of characterization of the obtained PEEK/CNTs composites, the chemical composition was not presented. More precisely, not a single method was applied in order to prove the chemical composition of the samples. It is necessary for the authors to add the CNOHS analysis of the CNTs sample, the analysis of the composition of polyether-ether-ketone, the analysis of the composition of all obtained PEEK/CNTs composites, as well as the XRD analysis of PEEK/CNTs composites in order to prove the degree of crystallinity that they constantly mention throughout the manuscript and thus connect all the characterized properties with crystallinity.
Our response: We thank the reviewer for pointing it out. The reviewer has proposed several excellent points that can be employed to enhace our claim, particularly on the experimental designs. For instance, the reviewer suggested to use XRD analysis to quantify degree of crystallinity in the obtained PEEK/CNTs composites. We believe this can provide a convincing validation on our findings and shall be included in future studies. In the current study, the crystallinity of the composites is quantified by the DSC, which is a common and widely accepted practice in similar studies, as stated in the following studies:
- Gohn, et al, Crystal nucleation in poly(ether ether ketone)/carbon nanotube nanocomposites at high and low supercooling of the melt, Polymer, 2020, 199, 122548. https://doi.org/10.1016/j.polymer.2020.122548
- Arif, et al, Multifunctional performance of carbon nanotubes and graphene nanoplatelets reinforced PEEK composites enabled via FFF additive manufacturing, Composites: Part B, 2020, 184, 107625.
https: //doi.org/10.1016/j.compositesb.2019.107625 - Gao, et al, Cooling rate influences in carbon fibre/PEEK composites. Part 1. Crystallinity and interface adhesion. Composites: Part A, 2000, 31, 517-530.
https://doi.org/10.1016/S1359-835X(00)00009-9
In addition, despite all mentioned good properties of the PEEK/CNTs composites, it is necessary to specify the concrete application of the obtained composites with emphasis on the scientific contribution. This type of examination, without clearly definition of the scientific contribution and mechanisms of influence of reactants PEEK/CNTs composites, represents the technical report. I kindly ask authors to add a scientific contribution.
Our response: We thank the reviewer for raising this point. In our manuscript, we have pointed out one potential application that stems from the present study, is for the electromagnetic shielding. The highly conductive CNTs have imparted the excellent electrical conductivity to the otherwise insulated PEEK matrix, which is highly desirable for the EMI shielding application. Our study further shows the annealing treatment, while significantly enhance the composites’ mechanical strength, can be adversary to the electrical and electromagnetic properties of the composites. The finding in this study thus opens broad opportunity to optimize the properties of the PEEK-CNTs composite through composition design and heat treatment.
I recommend the accept after minor revision (corrections to minor methodological errors and text editing)
Our response: We thank the reviewer for pointing this out. We have thoroughly proof-read the text and made corrections accordingly.
Round 2
Reviewer 1 Report
The authors have an important contribution in the preparation of dielectric materials, they have made modifications to their manuscript enriching their presentation, highlighting the important aspects of their research for which they accept the manuscript for publication.